# Third-Generation Heritage Spanish Acquisition and Socialization: Word Learning and Overheard Input in an L.A.-Based Mexican Family

**Eric Alvarez * and Aliyah Morgenstern ***

English Department, Faculty of Languages, Literature, and European Studies, Sorbonne Nouvelle University, 75005 Paris, France
* Correspondence: eric.alvarez.perez@gmail.com (E.A.); aliyah.morgenstern@sorbonne-nouvelle.fr (A.M.)

**Abstract:** This case study examines overheard speech in a third-generation heritage Spanish Mexican family. It presents Spanish use longitudinally and describes overheard Spanish word use in interaction. Transcribed on CLAN to create a plurilingual corpus, ethnographic video data consisted of 24 h across three sampling periods, yielding nearly 30,000 Spanish, English, and language mixed utterances. Quantitative analyses indicate strong Spanish use in the first sample, before dropping. Qualitative descriptions show the third-generation target-child's attunement to overheard Spanish, and her agency to use Spanish. Overheard input helps her use Spanish words, influencing her social encounters. This paper examines what we coded as overheard input in heritage language acquisition and socialization research. The language practices of one multigenerational Mexican family in California are explored, accounting for how their language practices in multiparty interaction co-create meaning, and how they help a third-generation child use Spanish words grounded in daily experiences. The findings contribute to the discussion of bilingualism in general and definitions of heritage bilingualism in particular. The results underscore the understudied role of overheard speech produced by a diversity of multigenerational family members and word learning. Participation frameworks are dynamically constructed by all participants as permeable, inclusive, and engage the children's use of inherited bilingual and bicultural practices, suggesting that heritage bilingualism is not just about abstract grammar.

**Keywords:** overheard input; heritage bilingualism; heritage Spanish; lexical development; multiparty framework; multigenerational interaction; multimodal analyses

## 1. Introduction

"Nearly two-thirds of the current mainland jurisdiction of the United States was, at one time, under a Spanish-Speaking sovereign-Spain or Mexico" (Macias 2001, pp. 334–35). While we will not dwell on the consequences per se of 400+ years of history, we will not forget it, since it brings needed context. This case study focuses on some social and linguistic aspects of how this ongoing tercera Hispanidad (Fuentes [1992] 2016), defined as the beating Hispanic pulse underlying some Anglo-American communities across the southern U.S., may influence the transmission of heritage Spanish acquisition and socialization across generations. Key demographic, linguistic, and educational data to further understand the conditions for intergenerational transmission will be presented in our methods section. Nevertheless, Spanish and English language use emerges in fronteras (borders) (Anzaldúa 1987), or contact zones (Pratt 1991), for example, in Los Angeles (henceforth L.A.), California, a place of linguistic and cultural convergence, or the local, physical

medium out of which culture, language, society, and consciousness get constructed. That construction (...) involves continuous negotiation among radically

heterogeneous groups whose separate historical trajectories have come to inter-sect; among radically heterogeneous systems of meaning that have been brought into contact by the encounter; and within the relations of radical inequality enforced by violence (Pratt 1996, p. 6).

In this type of hostile context, spread across space and time, heritage Spanish continually unfolds, shifts, and dies in some families and communities due to dynamic acquisition and socialization processes. But heritage Spanish is maintained in others. The language learning scenario for heritage Spanish bilinguals in L.A. also implies reduced input, and informal language learning opportunities partly due to limited or erratic bilingual language education policies in California (Rose 1989). This contact zone yields variable grammatical and interactional outcomes attested in adult heritage bilinguals (Montrul 2016), including heritage Spanish speakers (Valdés 2000) in the U.S.

Placed on a continuum, bilingualism is a moving target, showing great variation based on the breadth of unique bilingual experiences (DeLuca et al. 2019). Because language development is a life-long process (Nippold 2006), one may become more or less mono- or bilingual throughout ontogeny. While the term bilingual continuum refers to a range of bilingual profiles, here we also operationalize it to refer to the family's bilingual interactions or language modes (Grosjean 2008) as they fluidly use Spanish, English, or language mixing. Language mixing may include "words or morphemes from both their languages in one utterance" (De Houwer 2009, p. 41). Moreover, following Grosjean (2015), bilinguals may be in monolingual mode within language modes, i.e., interacting with speakers who they know only speak one of their languages. Or, bilinguals may be in bilingual mode, i.e., communicating with other bilinguals with whom the same languages are shared and with whom language mixing (bilingual talk) is accepted. We will see some of these language modes in the present paper, which in interaction fall along the bilingual continuum. Thus, the concept of bilingual continuum will be used to refer both to the linguistic interactions of the multigenerational family and their bilingual profiles. The figure below (Figure 1) illustrates some of the degrees of bilingualism. At the center, we find bilingualism and monolingualism at the extremities (Valdés and Figueroa 1994).

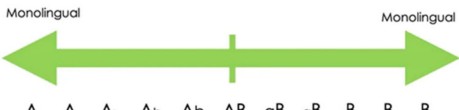

**Figure 1.** The bilingual continuum adapted from Valdés and Figueroa (1994, p. 8).

Moreover, most bilingualism studies have been experimentally grounded, frequently using monolingual tasks which may not reflect bilingual language interaction, for example, in creating metalinguistic representations (Torregrossa et al. 2022). Nevertheless, some researchers argue that experimental methods are not so different from naturalistic ones. Indeed, "Fieldwork essentially consists of tiny experiments that are fine-tuned in situ based on consultant feedback (. . .) there is no irreconcilable difference between fieldwork culture and the culture of laboratory linguistics" (Polinksy 2023, p. 1). Notwithstanding, a missing link is how heritage bilinguals go from being novice to expert bilingual speakers using naturalistic methods. Critical is understanding how heritage bilinguals' "acquisition (and socialization) could also be affected by social interaction and cognitive development" (Clark 2009, p. 2) through ethnographic approaches. Our longitudinal case study[1] (Alvarez 2023) asks the following question: *how, in a turbulent third-generation heritage Spanish acquisition and socialization context, does overheard Spanish input in a multigenerational environment impact a child's word learning in an L.A.-based Mexican family?* Our central question suggests a qualitative approach to the examination of specific occurrences of overheard (henceforth OH) input. The uniqueness and depth of the data analysis are highlighted by the inclusion of multiple generations and the variation in input depending on who is addressing the child. This includes the variety of language practices according to the family member addressing the child. A secondary question implies a quantitative approach to the input produced

by a variety of family members from different generations and with different levels of bilingualism. It seeks to determine the frequency of Spanish, English, and language mixing spoken longitudinally and within the environment over time, both at the individual and familial levels, as well as what the target-child's most robust semantic domains regarding Spanish word use are. It is critical to underscore that our quantitative analyses are purely descriptive in nature since ethnographic research is not concerned by inferential statistics. This aspect could be seen as a limitation to our study; however, future work may consider incorporating multivariate analyses. Nevertheless, in answering these research questions, we aim to fill three gaps in the research. First, the analyses of heritage bilingual development are grounded on the language socialization paradigm (Ochs and Schieffelin 2011), an "increasingly (used) framework" (Guardado 2018, p. 47) within heritage bilingual communities. This framework provides socio-culturally enlightened analyses of the continuity of heritage speakers' languages over developmental space and time (He 2006). These types of analyses also illuminate "how religious and heritage language institutions, along with familial units, support and amplify sociohistorically rooted language and cultural practices, attempting to draw children into an identification with a community of speakers" (Ochs and Schieffelin 2008, p. 11). Second, our analyses focus on the understudied role of overheard input produced by different family members in particular Spanish, but also English, and language mixed input. Third, this study considers an under-explored age group (3;10–4;9). Indeed, research on overheard input is rare, mainly focusing on the language development in children around 2 years old, for example, as seen in research by Foster and Hund (2011).

The acquisition and socialization into dynamic bilingual repertoires or linguistic forms and functions, along with the tightly interknitted, complex, and ever-transforming environmental dimension, engender a multiplicity of multigenerational heritage Spanish bilinguals. This paper delves into the understudied role of what is generally categorized as overheard Spanish input in the tension-ridden context of heritage language acquisition and socialization. By creating a plurilingual corpus, we explore the rich language practices of one multigenerational Mexican family belonging to a well-established population in multilingual and multicultural California. A plurilingual corpus may include several languages within the same text such as spontaneous bilingual interactions that highlight phenomena related to different types of language mixing (Léglise and Alby 2013). We show how the dynamic language practices in multiparty interaction support the meaning making process at the word level and how it helps a third-generation child use overheard heritage Spanish words.

We will first review the current state of the art on word learning, overheard input, and child agency while highlighting key work by an interdisciplinary group of researchers interested in child language development. A section on data and methods will follow. Since the data collection, transcription, and coding protocols are novel for such a bilingual setting, they will be described in detail. As this study applied a mixed methods approach, our analyses will be organized in two parts. Our quantitative analyses in part one consist of descriptive statistics to capture the family's linguistic soundscape, their individual language presentations, and the target-child's word-level semantic domain analyses. As an ethnographic study, part two will plunge us into three qualitative, or "thick" (Geertz 1973), descriptions of how the target-child uses overheard Spanish in mainly food orientated activities. Thick description is a key notion in Geertz's theory of culture where an ethnographer returns to the same data set and subsequently adds "layers" thereby rendering the analysis "thick". These analyses will be supported with plurilingual and screenshots to highlight their multimodal interactional nature. Then, our discussion will link the quantitative analyses per se with the qualitative ones, framing them around the family's often bilingual, typically multiparty, and certainly multigenerational ecology.

*Word Learning, Overheard Input, and Child Agency*

Monolingual experiences differ from bilingual ones and the unique language experiences of bilinguals may lead to divergent cognitive and social outcomes (Castro et al. 2022). Yet, most research has focused on monolingual children and their word-learning ability during adult–child interactions in which joint attentional frames (Tomasello and Farrar 1986) are constantly established. In these types of configurations, adult input is generally child-directed (Rowe 2008), and they are "created during social interaction that directs interlocutors' attention to some type of third entity (i.e., a triadic in which two or more interlocutors' attention is focused on concrete or abstract objects, contexts, and actions)" (Unger et al. 2015, p. 4). Joint attentional frames also inform children "how their language packages information at the word-level" (Clark 2009, p. 5). Interactional frames like these typically occur in Western middle-class communities where adults and young children teach toddlers new words through child-directed (henceforth CD) speech. But not all features of language, for example, how children learn to use pronouns (Oshima-Takane et al. 1999; Caët and Morgenstern 2015), are learned through speech addressed to children in joint interaction alone. Less didactic settings may also help children maintain the speed in which their vocabulary develops (Bloom 1998). Indeed, Clark (1996) and Hindmarsh and Pilnick (2002) advanced that the overhearer status in interaction had not been sufficiently studied. Interdisciplinary research interest thus emerged in the understudied role of OH input in language acquisition (Akhtar et al. 2001; Akhtar 2005; Floor and Akhtar 2006; Gampe et al. 2012). OH speech refers to verbal input that may be accessible but not necessarily directed to children in their language learning environment. The above-mentioned authors argue that OH input is also a vector for word learning, and that it may assist children's engagement in meaningful interaction. Moreover, since word-learning studies have largely focused on young children around 2 years old, researchers underscore "that it may also be important to investigate learning through overhearing in older children (…) Examining the language learning through overhearing in older children may provide us with a more complete picture of the language acquisition of this age group in general" (Boderé and Jaspaert 2016, pp. 1165–66). This makes the older child overhearer in or around an interaction illusive and particularly interesting. However, the stakes are further heightened when the input is OH in bilingual settings where two languages may be spoken separately or mixed. Studies of OH input in bilingual contexts may therefore help complete the picture of how bilingual language acquisition and socialization unfolds in naturalistic interaction or in language ecologies which are argued to be the worldwide norm (Harding-Esch and Riley 2003; Kroll and De Groot 2005; Daviault 2011; amongst others).

However, there exists great individual differences in language use and exposure. Children may either frequently overhear more than one language or have rare occasions to overhear input in a non-dominant language, such as heritage Spanish in some L.A.-based families. These occasions provide unique snapshots of the impact of OH speech on the acquisition and socialization of heritage Spanish words. As such, exposed to two languages, where each one may be used to various degrees in a sociolinguistic context where one is often dominant, typically developing bilingual children easily learn the words of at least one of their languages (De Houwer 2009). However, from a language acquisition and socialization perspective, we seek to understand how a third-generation child uses heritage Spanish, or her non-dominant minority language as it is spoken not only to her, but especially around her, i.e., when the speech is OH. Indeed, scholars have argued that overhearing plays a significant role in children's language learning process (Lieven 1994), especially in cultures where adults infrequently address them through CD speech. Moreover, in certain non-Western cultures where "communication is not child centered" (De León 2011, p. 84), OH speech makes up an important amount of linguistic input (Ochs and Schieffelin 2008). For example, recent research that quantitatively examined naturalistic bilingual input showed that children were exposed to important but variable proportions of OH and CD input in their bilingual language learning environments. This finding emerged in Orena et al.'s (2020) study that looked at how bilingual children in

Montréal, Canada, experience dual language input. Sperry et al.'s (2019) work also showed that children might experience OH input more abundantly than CD input, and that speech not addressed to children is therefore an important aspect of their language learning experience. Furthermore, some children grow up in large, multigenerational families where exposure to OH speech may be dramatically increased (Place and Hoff 2011; Shneidman et al. 2013; Soderstrom et al. 2018) due to a larger speech community. For example, in Los Angeles, California, many Latino families live in extended family configurations where grandparents, great-grandparents, aunts, uncles, and cousins, etc., may all share one roof (semi)permanently. This living arrangement is not necessarily typical in upper middle-class Euro-American families who tend to live in nuclear configurations consisting of only parents and children. Nevertheless, OH speech also makes up an important amount of input in non-middle-class family contexts (Sperry et al. 2018). Another result of multigenerational living styles among families of Mexican origin in L.A. is their constant use of Spanish and English (Bustamante-López 2008). Bi- or multilingual contexts like these further intensify patterns of language use and exposure at individual and family levels. Multilingual contexts are settings where multiple languages are used and may be described as harboring rich contextual linguistic diversity (Wigdorowitz et al. 2020, 2022).

The implications of OH input and word learning in relation to heritage bilingual development are manifold, especially since children show attentiveness to OH speech. Yet, another key concept that we underscore in our qualitative analyses and discussion, which is difficult to define according to Smith-Christmas (2021), is child agency. In a recent study, the notion of child agency is thoroughly reviewed and modeled within the emerging sub-field of Family Language Policy (Smith-Christmas 2021). However, in line with the field of linguistic anthropology, we discuss child agency within the language socialization paradigm. From this perspective, children are seen as actively co-constructing their socio-cultural and linguistic knowledge. That is, "Individuals (including young children) are viewed not as automatically internalizing others' views, but as selective and active participants in the process of constructing social worlds" (Schieffelin and Ochs 1986, p. 165). As a result of their choices and agency, children's status in interaction is dynamic, bidirectional, and even unpredictable as they constantly co-build webs of meaning. Moreover, child agency is embedded at the crossroads of input, interaction, context, and cognition. Cognitive-functional affordances and socio-interactional cues thus make language structure, meaning, and use easier for children to produce and apprehend. For example, Tomasello and Farrar (1986) insist that children are attentive not only to the interactional frames in which they find themselves, but also to social-intentional cues. Indeed, knowledge extends beyond an individual's mind. Knowledge also resides in the tools one uses, in the enabling environment, and in the collaborative efforts of multiple minds and bodies working together towards a common objective (Duranti 1997). Furthermore, because of their intention-reading skills used to disentangle the communicative functions and meanings of their languages, agency allows children and adults to continuously work together. The ability to discern intentions is likely a trait proper to humans, probably arising relatively recently in human evolution (Tomasello [1999] 2022). These skills are broad and not limited to linguistic communication. For example, intention-reading skills underpin various cultural abilities and practices routinely acquired, transformed, and resisted by children for example in using tools, or participating in rituals, and play (Tomasello 2003). A direct link between intention-reading skills and agency may therefore be established. They are anchored in joint interaction. That is, with an interlocutor, and his or her attention on a third entity. The role of child agency is non-negligible, and primordial throughout the language development process. Child agency, or the desire to engage in and subsequently shape interaction will be made explicit through our qualitative analyses. Children are movers and shakers in the bilingual, multimodal, and often multigenerational interactional processes in which they may be immersed. They are not just innocent bystanders of their linguistic worlds. Children are finely attuned to the social interactional frames as they assume varied roles such as ratified participants or overhearers (Goffman 1974, 1981), notions that are discussed further

below. Notwithstanding, "By examining the practices (...) of children in language minority communities (...) studies reveal how children (through their agency) take up, transform, or challenge the ways of speaking and thinking that are promoted through socialization practices" (Friedman 2011, p. 636).

Thus, from a usage-based approach to language acquisition and socialization (Morgenstern 2022; Morgenstern et al. 2021), we investigate how OH input in a dynamic bilingual language learning ecology motivates a third-generation child's use of Spanish words. This central question remains largely unanswered in the literature and this case study seeks to address it. Moreover, our case study, unlike few before it, presents multilevel, multimodal, and multiparty longitudinal analyses in which each semiotic resource is considered and in which several participant frameworks may overlap. The quantitative figures show the shifts in overall input, or how much Spanish, English, and language mixing are used at the family and the individual level. Nevertheless, some findings are attested in the literature, and our investigation contributes to this line of research. Next, following Geertz (1973), our "thick" qualitative analyses show how OH Spanish input in naturalistic bilingual interaction helps the target-child use words from her minority language Spanish, and how it helps her enter social encounters. Interactional descriptions of the like remain under-explored in heritage bilingual settings. We thus present a much-needed contribution in child bilingual language development research. Our cognitive-functional and social-interactional approach to analyzing OH Spanish words by a child disentangles "who speaks what language to whom?" (Fishman 1964). It accounts for each of the multigenerational interlocutors and the participation frameworks (Goffman 1974, 1981; Clark 1996; De León 1998, 2011; Morgenstern et al. 2021) that are constantly shifting and in which the third-generation child is exposed to OH Spanish words. Additionally, it considers the multimodal, or visual–gestural features such as gaze and pointing that facilitate the use of multiple languages (Benazzo and Morgenstern 2014, 2017) such as OH Spanish and English. Finally, the analyses track the parental responses (Lanza 1997; De Houwer and Nakamura 2021) to the child's use of OH Spanish words, assessing whether these often implicit responses promote a more mono- or a bilingual language learning environment where a bilingual child's use of OH Spanish input in the heritage language is nurtured.

## 2. Methods and Materials

### 2.1. Participants and Procedures

The naturalistic video-recordings used for the analyses mostly revolve around a little girl nicknamed LIN when she was 3;10 to 4;9. This paper focuses on the OH Spanish input in this L.A.-based, multigenerational family. In some recordings one can see and hear nearly a dozen adults who form an extended, multigenerational family. All the participants are on the bilingual continuum and mix languages to varying degrees. Based on our observations and experience with the family, the adults typically deploy three modes of communication: English, Spanish, or language mixing at home and in public, adjusting to the degree of bilingualism of their interlocuters. This paper thus focuses on word learning and OH Spanish input in the language acquisition and socialization process of a third-generation child.

The family tree below (Figure 2) was created using open-access software: https://www.familyecho.com/# (accessed on 20 November 2020). It charts the family's maternal organization. At the top, are the first-generation great-grandparents GRC and GRT. At the bottom, we find their four third-generation great-grandchildren. The three-letter uppercase codes identify each speaker. The number to the right of the three-letter code indicates each speaker's generation. We will expand on the generation code further below. The color scheme represents their biological gender. The red circle designates LIN our target-child. The pink colored box indicates that she is female, and the number 3 next to the three-letter code indicates her generation.

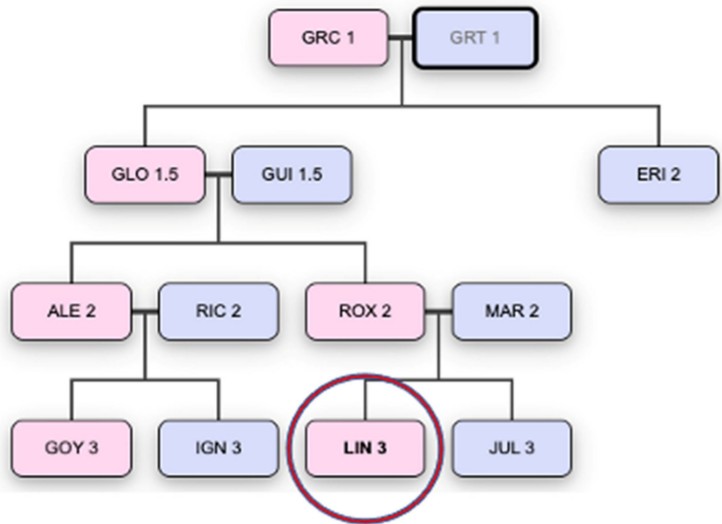

**Figure 2.** Three-generation family tree and the maternal line.

Considering the potential impact of OH input on LIN's language development, here we provide additional sociolinguistic details about her language exposure and use. Indeed, contextual details about her bilingual ecology, such as entry in school, whether she is a simultaneous or sequential bilingual, etc., are crucial. These environmental aspects may provide an enhanced understanding of the results presented in the descriptive statistics, as well as the thick descriptions in the following two sections. LIN, a simultaneous bilingual, was 3;10 at the time of recording and had not started formal schooling in English. She is the first-born daughter of ROX and MAR. LIN was born and is being raised in South L.A., benefiting from varying degrees of English, Spanish, and mixed language input. Nevertheless, in Alvarez (2023), an examination of data of CD input shows that overall English is used the most when the adults speak with LIN. The same is true when the women address each other. Moreover, Alvarez's (2020) results peered into her language use when she was 1;10, showing that English was the most frequently used language. Therefore, it is no surprise if today English also accounts for the largest proportion of her utterances[2]. LIN used English at almost 57%. There were slightly over 22% of Spanish utterances and less than 1% of them were mixed. LIN thus uses English the most in terms of frequency. It also matched that of her environmental input, and it was followed by Spanish. The proportion of her language mixing was not quite 1%, placing language mixing as her third mode of communication. Moreover, LIN is also growing up with three other children: GOY her older cousin, IGN her younger cousin (both the children of ALE and RIC) and JUL, her younger brother. Of the group, LIN is the second eldest. In this scenario, except for LIN, the other three children are going through atypical language development. This social context raises an issue that will not be the object of inquiry at present: how does growing up with atypically developing bilingual children help or hinder bilingual linguistic development?

Moreover, the data consist of 24 h of video-recordings collected over the course of one year (2018–2019). The video-recordings were made inside and outside of two homes owned by the 1.5-generation grandparents (GLO and GUI). The concept of 1.5-generation will be expanded on below. The homes sit next to each other on two separate lots. There is always a flow of family members going to and from their homes. The video-recordings took place mainly during mealtime, but also while playing in the garden. Even if the family lives in L.A., the Spanish and Mexican cultural and linguistic legacy is strong in this area like in much of California (Macias 2001). The family established themselves in Florence-Firestone in the late 70's. The table below (Table 1) synthesizes the most recent demo-linguistic data of the community to better apprehend the conditions for heritage Spanish transmission.

**Table 1.** Community characteristics adapted from Figures 2 and 3 and Tables 4, 5, and 7 in L.A. County Department of Regional Planning (2019).

| Data Type | Florence-Firestone Population Characteristics | Percentage | Year |
|---|---|---|---|
| Population | 65 752 | not applicable (henceforth na) | 2019 |
| Age | <9 | 6% | 2019 |
| | 10–14 | 9% | |
| | 15–19 | 9% | |
| | 20–64 | 57% | |
| Ethnicity | Hispanic | 90.73% | |
| Language use | speak Spanish | 86% | 2019 |
| | speak Spanish and English | 63.43% | |
| | speak English not well | 19.50% | |
| | Speak no English | 17.07% | |
| Educational Outcomes (out of 34 363) | <than 9th Grade | 38.08% | 2019 |
| | High School Diploma | 22.65% | |
| | Some College/No Degree | 11.66% | |
| | Bachelor's Degree | 3.06% | |
| | Grad/Professional Degree | 1.19% | |

Florence-Firestone is nearly a 66 thousand strong unincorporated community in South L.A. A total of 57% of the population is between 20 and 64 years of age. Immigration is high, coming from all over Latin America such that just around 90% of the community members are of Hispanic origin and mainly from Mexico. At 86%, most community members speak Spanish, nearly 65% are bilingual, and almost 20% do not speak English. Educational attainment is low. Nearly 40% have attained less than a 9th grade education. The video-recordings are thus anchored in a highly bilingual heritage Spanish–English contact zone where demo-linguistic factors play an important role in heritage Spanish transmission. Indeed, Spanish is typically acquired orally, but language mixing is also frequent. Finally, Spanish is both shifting and being maintained.

*2.2. Data Collection*

Capturing the process of heritage language acquisition and socialization challenges linguists interested in analyzing naturalistic interaction. A critical issue is documenting dynamic language practices when "individual variability is a hallmark of heritage speaker groups" (Montrul 2016, p. 165). Our study addresses this issue by presenting extensive quantitative data in the form of language presentations of the five central participants. Following the linguistic anthropological approach, naturally occurring speech was recorded ethnographically (Ochs and Schieffelin 2011) by the first author and family member. As such, participation in the interactions which may be viewed as not traditional was non-negotiable since researcher participation could significantly bias the results. This challenge and the steps to maintain objectivity as much as possible are discussed in Alvarez (2023). However, following Williams (2008) and Morgenstern (2009), an observer's (i.e., a stranger's) impact onsite is not a major hurdle in the collection of spontaneous data. Therefore, a family member's impact who must participate in interaction may be even less influential onsite. Indeed, "It seems that the ordinary living of life in family contexts soon displaces awareness of recording, so recording interaction between caregivers and children over long, uninterrupted stretches of time does provide authentic data" (Williams 2008, p. 67).

Thus, to collect the data we used a Canon HD Legria HF G25 camcorder, four SanDisk Ultra 128 GB microSDXC-I memory cards, and a Sunpak Ultra 6000PG tripod. The video-

recordings were stored and backed up in a WD 4TB My Passport for Mac portable external hard drive. Analyses of natural interaction is also advantageous in moving the field of heritage linguistics forward (Polinsky and Scontras 2019). Critically, collecting spontaneous language data requires substantial human and material investment despite the object of study, and the length of fieldwork required to collect it (Tomasello and Stahl 2004). However, longitudinal case studies allow to verify fluctuations in language use, and development across the lifespan. Thus, to chart heritage bilinguals' language development over time, "the ideal research design (. . .) would be a longitudinal study following the same individuals and documenting changes in their linguistic behavior from time 1 to time 2 or time n" (Montrul 2016, p. 168). Moreover, prominent researchers (De Houwer 2009; Wei 2000; Pauwels 2016) recommend having advanced knowledge of the languages and cultures involved. Indeed, bilinguals' linguistic behavior may only be adequately apprehended with a certain degree of insider knowledge of the community where said behaviors are displayed, and with an understanding of the circumstances that lead to such behavior (Gardner-Chloros 2009). As a community member, the first author meets this recommendation. Nevertheless, he was not always present in each video-recording for reasons related to the very act of being social. Thus, at times the camera was intentionally left rolling. However, he was present and participating in most of the interactions. When this is not the case, it is mentioned at the onset of the analysis. Finally, participation in the ebb and flow of pre-established family life requires harmonious integration into the family's complex bilingual and sociocultural structure.

### 2.3. Data Analysis and Coding

The data were first transcribed using the open-access: https://dali.talkbank.org/clan/ (accessed on 20 November 2020) software CLAN, version number V 20-Nov-2020 11:00. CLAN stands for Computerized Language Analysis (MacWhinney 2000) and it was regularly accessed between 2020 and 2022, following the CHAT coding conventions. The main tools used throughout the transcription phase included three devices of Apple hardware: a Macbook Pro 13" 2,3 Ghz Intel Corei5, a Magic Keyboard, and a Magic Mouse 2, as well as Bose QuietComfort 35 Wireless Headphones II and an Essentielb Pixel Clear 24VH PC Screen. Following the language socialization paradigm, transcription was viewed as providing an analytical path that could lead to nuanced understandings of how members of society engage in interaction. This process yielded a plurilingual corpus which are still rare today. While second-by-second transcription served as the first level of analysis, the data were then exported to Excel for further coding. Coding is an analytically driven form of data organization that may vary according to a researcher's agenda (Heller et al. 2018). Following Ochs (1979), coding was kept simple in both quantity and quality. In total, 14 sociolinguistic categories were further coded. The table below (Table 2) lists 12 of the coding categories in column one [3], describes them in column two, and finally groups them into the type of data in column three (we do not use all the results of our coding in this specific article; see Alvarez (2023) for a full presentation of all the analyses conducted on the data).

A total of 12 coding categories were considered, and several measures were taken using Excel's pivot tables, or cross-tabulations. Our methodology allows us to generate quantitative measures of LIN's bilingual ecology which are generally unaccounted for in studies of bilingual language acquisition. These measures also allow us to focus on qualitative multigenerational, multimodal analyses of heritage language socialization in diverse, and ever-shifting multiparty participation frameworks. Our mixed methods approach following Stivers (2015) and Beaupoil-Hourdel and Morgenstern (2021) thus allows us to consider links between interactional practices (individual, familial, and environmental) and sociolinguistic variables. Three examples are presented and briefly discussed further below. The three "thick" descriptions to follow based on the Spanish OH words include: "tortilla" (flatbread), "mosca" (fly), and "chile" (chili). These extracts were selected using the Speaker code and Language code which made LIN's Spanish output easier to analyze.

Table 2. The 12 social and linguistic coding categories.

| Coding Categories | Description | Type of Data |
|---|---|---|
| Language | To identify the language(s) of the input/output under study: English (E) and Spanish (S). A language [- mix] code was not used on CLAN when transcribing, and language mixing (when words or morphemes from more than one language are integrated into an utterance by a speaker) was coded on Excel using the code for language mixing (M). The first two codes overlap with the CHAT coding conventions but used in Excel along with the code for mixing adds possibilities for quantitative analysis. | Metadata |
| Input type | To determine if the input to which the target-child LIN reacts is child-directed or overheard. CLAN allowed to code for input type by using the CHAT code %exp: to insert a comment beneath a turn, Excel formalizes this code, and increases its analytical potential since these instances became quantifiable. | |
| Receptive comprehension | To tag when LIN seemed to understand child-directed Spanish and was thus developing a receptive mode of communication. A demonstration of LIN's receptive Spanish comprehension was generally marked by her contextually appropriate response, oral or gestural, in interaction. The CHAT code %exp: allowed the researcher to insert a comment after a turn, however on Excel it was formalized, thus its analytical potential is increased since these instances became quantifiable. | Linguistic data: to analyze linguistic variables |
| Language of previous utterance | Allows the researcher to establish the language of the previous utterance: English (E), Spanish (S), mixed (M) to see impact on speakers' output or language choice in interaction. | |
| Word categories (Nelson 1973) | To determine if the target-child LIN's heritage Spanish can be classified in sub-groups (equivalent to semantic domains), i.e., cultural words, food, kinship etc. This code may help us consider Grosjean's (2015) Principle of Complementarity and build a semantic domain analysis. | |
| Transcription | To stabilize and analyze the written record of oral data. The transcript includes data in American English, Mexican-based Spanish, and language mixing at the discourse level. | |
| Speaker | To identify the speaker whose Spanish, English, or language mixing was transcribed. | |
| Generation | To identify the speakers' generation: first- (1G) = immigration to the U.S. as adults (age 18+), first-and-a-half- (1.5G) = immigration to the U.S. during middle childhood (age 6–12). Organizing "late" arrivers in terms of generation is a fuzzy issue. One proposal is to consider "late", non-adult arrivers to be between the first- and second-generation (Faez 2012). This is the position assumed here. Next, second- (2G) = native-born children of immigrant parents, and finally, third-generation (3G) = native-born children of native-born parents. | |
| Sex | To identify a speaker's biological aspects as determined by their anatomy as assigned at birth. The participants coded as female (F), or male (M). | Social data: to analyze social variables |
| Addressee | To identify who the speaker was addressing in interaction. This study was restricted to productions addressed to one main participant. | |
| Context | Allows the researcher to determine whether linguistic phenomena occurred in specific locations, i.e., in the kitchen, living room, dining room, in the front yard etc., while doing certain activities, i.e., eating, playing etc., or while discussing certain topics, i.e., family, food, education, money etc. | |
| Motivation/source | To interpret what caused the speaker to produce a particular utterance. | |

## 3. Results: Descriptive Statistics

### 3.1. Linguistic Soundscape

Excel allowed to represent the selected codes above in terms of absolute and relative figures in the table below (Table 3). This enabled us to capture LIN's linguistic soundscape, i.e., the longitudinal input fluctuations of OH and CD input in her ecology. Indeed, the consistent and continued use of heritage languages may inform developmental outcomes in heritage bilinguals (Caloi and Torregrossa 2021). Spanish (yellow column) abruptly drops from 50% before stabilizing around 19% and 22%. English (blue) jumps from 43% and then stabilizes around 72% and 69%. Language mixing (green) remains low and stable, first at 7% before rising to 9% in the last two sampling periods. The frequencies closely match LIN's own mastery of English. The increased and sustained use of English may reveal how the adults switch to her dominant language. LIN as a novice speaker thus seems to exert her influence on the more experienced multigenerational and bilingual family members.

**Table 3.** Linguistic soundscape evolution of OH and CD speech: February 2018 to January 2019.

| Month | Spanish | | English | | Mixed | | Total |
|---|---|---|---|---|---|---|---|
| | Absolute | Relative | Absolute | Relative | Absolute | Relative | |
| February 2018 | 4 266 | 50% | 3 638 | 43% | 563 | 7% | **8 467** |
| August 2018 | 1 138 | 19% | 4 246 | 72% | 499 | 9% | **5 883** |
| January 2019 | 2 863 | 22% | 9 067 | 69% | 1 238 | 9% | **13 168** |
| Total | 8 267 | | 16 951 | | 2 300 | | 27 518 |

The table highlights the omnipresence of English (16,951 utterances) longitudinally. It is the most used language in the family. Spanish is used the second-most spoken (8267 utterances). Finally, language mixing never represents more utterances than unilingual English or Spanish utterances. Language mixing (2300 utterances) represents the third mode of communication multigenerationally. However, we may consider a key factor impacting the family's heightened use of English and decreased use of Spanish across the three samples. For example, a strong variable is GRC's near absence after the first recording period (February 2018). GRC, a first-generation participant, is LIN's great-grandmother, and she speaks Spanish nearly 100% of the time. During the first sample, she came from Mexico to spend the month with ERI (her son) as he carried out his data collection. GRC's presence thus seems to boost the family's use of Spanish. However, GRC does not participate in the second sample (August 2018) and hardly participates in the last one (January 2019). GRC's presence therefore encourages a stable and strong use of Spanish across family members. Moreover, when GRC is absent, the family seems to adjust, or shift to English, with some Spanish use and language mixing of course. As such, the presence of family members who predominantly speak Spanish with the other family members could potentially influence the language dynamics or how much each language is used longitudinally, and thus LIN's exposure to heritage Spanish.

*3.2. Language Presentations*

A heritage language socialization environment emerges out of the individual bilingual practices or language presentations within a multigenerational family. Indeed, "Depending on the specific family and culture they are born into, babies will have many people to interact with or very few" (De Houwer 2009, p. 99). The interlocutors below form part of LIN's tight-knit speech community where the adults engage in the situated child-rearing process (Lave and Wenger 1991). We focus on the language presentations of LIN's primary caretakers, the women on the maternal side: ROX (mother), ALE (aunt), GLO (grandmother), and GRC (great-grandmother).

Table 4 below shows how the speakers in LIN's family use Spanish, English, and language mixing. For each of the five speakers, including LIN, the analyses give the absolute number of utterances and their relative frequency longitudinally. Moreover, to visually capture both the stability and variability in their language presentations, we begin with the speaker with the most consistent language use pattern and end with the speaker demonstrating the most variability. We therefore start with the target-child LIN who uses Spanish, English, and language mixing the most consistently. LIN is followed by GRC whose language presentation is also stable. Increased variability emerges in language use with ALE and ROX (sisters) who share similar language presentations longitudinally. The relative frequencies with which ALE and ROX use Spanish, English, and language mixing do not show increasing or decreasing trends, but rather display an erratic up-down-up pattern or the reverse across languages. Finally, GLO presents the most variability, but it is not erratic like that of ROX and ALE. In Spanish, GLO shows a steady decreasing trend, while for English and language mixing the trend is to increase. Nevertheless, the results in the blue column above reveal a strong preference for the use of English for almost all the speakers even if this bilingual ecology lends itself to social encounters where the language of conversation may be entirely in Spanish, especially in the presence of GRC (1G), as discussed further up. Finally, an additional comment to be made is related to generation (column three) and language presentation. The speakers presenting the most stability

in language use hail from the first- (GRC) and third-generation (LIN), while those that present the most variability come from the second- (ROX and ALE) and the 1.5-generation (GLO). Indeed, ROX, ALE, and GLO according to the bilingual continuum presented in the introduction are the 'most' bilingual and therefore have 'more' linguistic resources at their disposal which in turn may heighten their variability in language use.

**Table 4.** Relative frequency: language presentation of Spanish, English, and mixed utterances.

| Month | Speaker | Generation | Absolute (Total) | Spanish | | English | | Mixed | |
|---|---|---|---|---|---|---|---|---|---|
| | | | | Absolute | Relative | Absolute | Relative | Absolute | Relative |
| February 2018 | | | 1 273 | 44 | 3% | 1 155 | 91% | 74 | 6% |
| August 2018 | LIN | 3 | 846 | 39 | 5% | 772 | 91% | 35 | 4% |
| January 2019 | | | 1 803 | 33 | 2% | 1 714 | 95% | 56 | 3% |
| February 2018 | | | 1 714 | 1 695 | 99% | 3 | 0% | 16 | 1% |
| August 2018 | GRC | 1 | na | na | na | na | na | na | na |
| January 2019 | | | 586 | 559 | 95% | 5 | 1% | 22 | 4% |
| February 2018 | | | 190 | 10 | 5% | 163 | 86% | 17 | 9% |
| August 2018 | ALE | 2 | 812 | 108 | 13% | 613 | 76% | 91 | 11% |
| January 2019 | | | 780 | 63 | 8% | 638 | 82% | 79 | 10% |
| February 2018 | | | 1 874 | 733 | 39% | 948 | 51% | 193 | 10% |
| August 2018 | ROX | 2 | 1 101 | 87 | 8% | 940 | 85% | 74 | 7% |
| January 2019 | | | 2 849 | 340 | 12% | 2 271 | 80% | 238 | 8% |
| February 2018 | | | 644 | 361 | 56% | 191 | 30% | 92 | 14% |
| August 2018 | GLO | 1.5 | 786 | 302 | 39% | 379 | 48% | 105 | 13% |
| January 2019 | | | 2 215 | 571 | 26% | 1 217 | 55% | 427 | 19% |

*3.3. Semantic Domains: Heritage Spanish*

Longitudinally, LIN uses 96 Spanish words (types) distributed into roughly 19 semantic domains or specific areas of cultural significance (Ottenheimer 2005). The top five semantic domains for LIN include "Food" (24 types), "People" (11 types), "Numbers" (8 types), "Objects" (8 types), and "Motion" (7 types). The three-column table below (Table 5) presents them according to the 19 semantic domains used to organize LIN's heritage Spanish repertoire.

**Table 5.** The 19 semantic domains for heritage Spanish.

| Number of Semantic Categories | Semantic Category | Number of Examples in Data |
|---|---|---|
| 1. | Food | 24 |
| 2. | People | 11 |
| 3. | Numbers | 8 |
| 4. | Objects | 8 |
| 5. | Motion | 7 |
| 6. | Animals | 6 |
| 7. | Sense perception | 5 |
| 8. | The body | 4 |
| 9. | Colors | 3 |
| 10. | The physical world | 3 |
| 11. | Clothing and grooming | 3 |
| 12. | Adverb | 2 |
| 13. | Expressions | 2 |
| 14. | Spatial relations | 2 |
| 15. | Emotions and values | 2 |
| 16. | The house | 1 |
| 17. | Time | 1 |
| 18. | Speech and language | 1 |
| 19. | Quantity | 1 |

About a quarter of the Spanish words that LIN uses are food related. Based on the semantic domain analysis, on the language presentations, and on the linguistic soundscape analyses, we gain a deeper understanding of the dynamic language practices. We see how the linguistic resources are used by the family and by the individuals, down to the distinct Spanish words that LIN produces, showing both the importance of food and the impact of culture. Of course, it is crucial to keep in mind that these results may be a direct influence of the specific context or the data collection method which predominantly took place in the kitchen and in the dining room during mealtime. Nevertheless, the next three thick analyses reveal how OH Spanish input, despite the increasing use of English, helps LIN use OH Spanish words in her bilingual and bicultural context, an environment anchored in rich multigenerational and multiparty interactions. Finally, the present overview of the family's language practices is essential, informing the qualitative analyses below and the ensuing discussion.

## 4. Results: "Thick" Descriptions of Overheard Spanish—Food for Thought, and Other Bits

In Alvarez (2023), we examine LIN's use of heritage Spanish in interaction in and around the kitchen before, during, or after mealtime activities. In these analyses, emphasis was placed on the provision of CD input by various family members within ever-shifting participation frameworks. However, it is critical to underscore two issues as we move forward. First, to offer richer insight into LIN's OH Spanish word use, future studies should include additional interactional contexts beyond meals, thus minimizing the chance of our results being unduly swayed by specific contexts. Second, we highlight the intended thematic similarity in two of the thick descriptions, i.e., "tortilla" (flatbread) and "chile" (chili). Both food-related words have been borrowed into English from Spanish, so we may wonder if indeed they are truly Spanish in these instances. Given the interactional context and the phonological contours of their pronunciation, we argue that there is no ambiguity that "tortilla" (flatbread) and "chile" (chili) are produced in Spanish. These precise moments will be briefly developed in each of the corresponding qualitative analyses. Nevertheless, in the thick descriptions to follow the adults are all bilingual speakers, spanning all three generations. A result of this multigenerational environment is that a broad range of linguistic resources are deployed in multimodal interaction that span the bilingual continuum. The following three analyses focus on how LIN uses OH input in her bilingual ecology to produce Spanish words. OH speech also allows her to enter the interactional frame in progress and to encourage other novice speakers to do the same.

### 4.1. Tortilla

In the exchange below (Box 1), LIN overhears ROX tell her little brother JUL that he has a "tortilla" (flatbread) in his hand. A "tortilla" (flatbread) is round, thin, and made of corn. It is consumed hot and commonly accompanies meals in Mexican households[4]. The interaction takes place in February 2018. GRC, JUL, ROX, GRT, and LIN are sitting around the dinner table. ERI (the observer) is not visible and is sitting behind the camera placed on the table near him. We can see most of the other participants. JUL is partially visible since even if he is sitting on GRC's lap, his face is behind two large honey- and olive oil-filled bottles. Therefore, gaze and gestures are analyzed. The language of interaction is almost entirely in Spanish. ERI notices JUL manipulating a piece of "tortilla" (flatbread). ERI comments to the family how using this "tortilla" (flatbread) as a tool to eat is culturally learned both from watching others and by practicing himself. The discussion is unfocused and there is no sense of distraction since there is no objective other than talking casually and observing JUL eat. Finally, only the adult speakers in the participation framework are ratified, i.e., addressed through directed speech in the interactional frame. JUL and LIN are unratified, i.e., unaddressed, and therefore overhearers. However, the interaction shows LIN's attentively interpreting the environmental input.

**Box 1.** LIN is 3;10 (*Plurilingual Transcript 1; ex. 19: FEB_21_2018_ROX_ERI_GRC_GRT_JUL_LIN_tortilla*).

1. *ERI:   [- spa] es cultural aprender +…
              It's cultural to learn +…
2. *ERI:   [- mix] comer con la tortilla como los Chinos que comen con los chopsticks@s +…
              To eat with tortillas like the Chinese who eat with chopsticks +…
3. *ROX:   The +… @ERI
4. *GRT:   [- spa] uhum. @ERI
              Uhum. (Gazes at ERI and agrees, slightly nodding his head)
5. *ERI:   [- spa] con los las los palitos +…
              With the little sticks +…
6. *GRC:   [- spa] es su cultura. @ERI
              It's their culture. (Lifts her head, and looks at ERI in agreement)
7. *ERI:   [- spa] cultura también entonces se me hace simplemente increíble de que pues +…
              Also, their culture so it seems incredible to me that +…
8. *ROX:   [- spa] es chile Papa [: Papá]. @JUL
              It's chili Dad.
9. *JUL:    what? @ROX
10. *ROX:   [- spa] chile. @JUL
              Chili
11. *JUL:    xxx. @ROX
12. *GRT:   [- spa] pinche mocoso mocoso. @JUL
              Little booger boy.
13. *GRC:   [- spa] dame un toallita. @ERI
              Give me a napkin.
14. *GRT:   [- spa] toma toma. @GRC
              Take it take it.
15. *JUL:    more.
              (Points to the "tortilla" (flatbread) basket)
16. *GRT:   [- spa] mocoso mocoso! @JUL
              Booger boy.
17. *GRT:   [- spa] pinche mocoso mocoso. @JUL
              Little booger boy.
18. *JUL:    xxx.
19. *GRC:   [- spa] qué qué pasa? @JUL
              What what's happening?
20. *ROX:   [- spa] la cara.
              The face.
21. *LIN:    I told you he xxx.
              (Looks at JUL)
22. *ROX:   [- spa] que hace nomás por payaso.
              That he makes just to be silly.
23. *ERI:   [- spa] sí por eso entonces digo +…
              Yes, that's why I say +…
24. *JUL:    more.
              (leans towards ROX)
25. *ROX:   [- spa] tortilla. @JUL
              Tortilla. (Looks at JUL)
26. *ERI:   [- spa] cree que le +…
              Thinks that +…
27. *LIN:    [- spa] tortilla.
              Tortilla.
28. *ROX:   [- spa] tortilla. @LIN
              Tortilla. (Glances at ERI, then LIN, then GRT)
29. *LIN:    [- spa] tortilla. @ROX
              Tortilla.

ERI draws the family's attention to JUL who is busy eating with a "tortilla" (flatbread). They are thus in multiparty interaction. In Spanish, ERI (l.1) states that to eat with this type of "tortilla" (flatbread) is cultural. Then, through language mixing, ERI (l.2) suggests that this may be like how Asian children learn to use chopsticks, a culturally unique way of

consuming their meals. Ochs et al. (1996) explored the dinnertime activities of 20 middle-class families both in the U.S. and Italy. In their comparison of the socialization of "taste", they found that a critical site for the socialization of culturally specific eating habits (among others) is around negotiations over food. While food was not the object of negotiation here, JUL imitated the culturally specific way of eating in this Mexican-origin household. ROX brings her gaze back to JUL. GRT is ratified and listening to ERI but looking at JUL, grinning and sitting across from JUL. As ERI finishes his mixed language utterance, GRT gazes at ERI and agrees (l.4), slightly nodding his head. GRC who is also a ratified speaker, but focused on her meal, lifts her head and looks at ERI to agree in Spanish that it is indeed cultural (l.6). GRT and GRC's acceptance of ERI's proposition aligns with what Blum-Kulka (2002) and Keppler and Luckmann (1991) have argued: that family reunions may lead to teaching moments where a knowledgeable self-appointed member (ERI) attempts to provide an explanation of either social or natural phenomena. Family gatherings like these are therefore critical sites for the social construction of knowledge (Kiaer 2023). In the meantime, LIN who is unratified sits on ROX's lap. Her attention seems to be focused on a black pen that she is manipulating with her hands. It is not clear whether she is paying attention to the OH speech. However, JUL (l.15) seems to point to the "tortilla" (flatbread) basket, but instead gets his nose wiped. While ROX responds in Spanish (l.20) to GRC to say that it was funny that JUL made a funny face after getting his nose cleaned, LIN (l.21) looks at JUL, suggesting her receptive comprehension of Spanish and that she is indeed attentive to OH speech. This may be corroborated by the fact that in ratifying herself, LIN utters something that begins with "I told you he +...", but the remainder is undiscernible. It is also not clear who LIN addresses, but a strong guess is ROX. What is certain, however, is that even if LIN's attention seems to be focused on the black pen, she is attuned to the OH Spanish input in her environment, understanding to an extent what is being said. We therefore discern a sense of her receptive bilingualism. However, LIN's commentary is unacknowledged by the adults in the multigenerational and multiparty participation framework. Since she does not manage to break into the interaction, she continues to play with the black pen. Next, JUL (l.24) in English, seems to say more as he leans towards ROX. ROX's reaction in Spanish (l.25) confirms that JUL has a "tortilla" (flatbread) as she looks at him and seems to take it from his hand. ERI (l.26) continues in Spanish but stops to see if JUL will repeat the Spanish word that ROX models for him. JUL does not repeat ROX's CD Spanish input. On the other hand, LIN overhears ROX and captures her attention. LIN stops playing with the black pen and observes the "tortilla" (flatbread) being passed from JUL to ROX. Next, LIN (l.27) without ratifying herself into the conversation repeats the OH Spanish word "tortilla" (flatbread) with the relevant Spanish phonology[5], but with no clear addressee, after which she goes back to her task. Then, ROX who up to this point is focused on JUL, but also glances at ERI, turns to LIN realizing that LIN repeated the OH Spanish. ROX's glance then goes from LIN to GRT who following LIN's repetition draws GRT's attention to her. ROX (l.28) therefore intentionally remodels the OH Spanish word for LIN, perpetuating the bidirectional nature of spontaneous interactions (King and Fogle 2013). LIN (l.29) recognizes that ROX's Spanish utterance is meant for her. LIN thus briefly looks at ROX and then, as she looks back down, softly repeats "tortilla" (flatbread), in what we could consider as Spanish thanks to its phonological specific features, one more time with a sort of smile on her face. This is perhaps because she was 'caught' practicing heritage Spanish on her own. LIN's use of OH Spanish ends there and the bilingual adults continue with their unfocused conversation in their multigenerational and multiparty framework.

This sequence illustrates how even though LIN's visual attention seems to be focused on her object manipulation, her multimodal behavior indicates her dynamic reception of the OH speech which is then demonstrated by her own verbal productions in Spanish. Our detailed analysis confirms the importance of multiparty interactions at the dinner table in bilingual language socialization and the impact of a culturally specific semantic domain—food.

*4.2. Mosca*

In the next example (Box 2), LIN is an unratified speaker and playing alone in the living room. She is removed from the multigenerational dyadic participation framework in the dining room. Moreover, despite the spatial distance between the two areas in the house, LIN, through her engagement in the interaction, shows her attunement to OH Spanish. Like the example above, the OH Spanish is addressed to her brother JUL in a joint interactional frame. Both adults are bilingual speakers, and they represent the 1.5-generation (GLO) and the second-generation (ERI). In this brief social encounter, the adults are speaking Spanish and English. Through the following analysis, we underscore how LIN overhears and then uses the Spanish input afforded by her bilingual ecology. In contrast to the semantic domain of food, there is an English term for "mosca" (fly) that LIN does not use here. As LIN overhears GLO say the word "mosca" in Spanish to JUL, not only does she repeat it, but she also encourages him to do the same. The present interaction was recorded in February 2018. GLO, JUL, and ERI are sitting around the dinner table after dinner. ERI is behind the camera, which is facing GLO and JUL. LIN is not visible until she peers her head from behind the couch in the living room. Since it is after dinner and almost time for bed, the lights are dimmed. However, visibility is good enough to consider gaze and gestures. Finally, the slow-paced conversation is unfocused, so there is no real sense of distraction. Only LIN is the unratified overhearer until she captures the OH Spanish word from afar.

**Box 2.** LIN is 3;10 (*Plurilingual Transcript 2; ex. 20: FEB_15_GLO_JUL_LIN_ERI_2018_mosca*).

| |
|---|
| 1. *GLO:   [- spa] mosca. @JUL |
|                            Fly. (Looks at JUL) |
| 2. *GLO:   [- spa] mosca. @JUL |
|                            Fly |
| 3. *JUL:    xxx. @GLO |
|                       (Looking and pointing right finger in fly's direction) |
| 4. *GLO:   [- spa] mosca. |
|                        Fly. (Elongates vowel while in joint attention with JUL, then looks where JUL is pointing) |
| 5. *ERI:    what are you doing LIN? |
| 6. *LIN:    [- spa] mosca. |
|              Fly. (Peers up from behind the couch) |
| 7. *LIN:     Mickey Mouse. @ERI |
| 8. *LIN:    [- spa] mosca? |
|                            Fly? (Rising intonation) |
| 9. *GLO:   [- spa] es una mosca @JUL |
|                        It's a fly. |
| 10. *LIN:    [- mix] say mosca@s Julian. |
|                        Say fly Julian. |
| 11. *JUL:    xxx. @GLO |
| 12. *GLO:   [- spa] ya se fue. @JUL |
|                        It's gone. |
| 13. *GLO:   [- mix] anda una mosca por allí he's@s all@s amused@s con la mosca you@s see@s it? @ERI |
|                        There's are fly around here he's all amused with the fly you see it? |
| 14. *ERI:    is he? @GLO |
| 15. *GLO:   [- spa] aie hijo de tu madre. @JUL |
|                        Son of a buck. |
| 16. *GLO:   [- spa] pinchi [: pinche] panzón. @JUL |
|                        Damn fatass. |
| 17. *JUL:    xxx. @GLO |

JUL is sitting on GLO's laps, positioned such that they are face-to-face. He is captivated by a fly that has been buzzing around. Responding to his curiosity, GLO (l.1) looks at him and casually says "mosca" (fly), establishing a dyad in joint interaction. A second goes by before JUL utters unintelligible speech (l.3). He is looking and pointing with his right

finger in the direction of the fly in the kitchen. His multimodal production, visual, gestural, and vocal, which likely support him in coordinating his social interactions during this developmental phase, is executed while GLO (l.4) repeats the word "mosca" (fly). They are therefore still in joint attention and focused on a third object. However, this second time around GLO elongates the vowel as she first looks at JUL and then looks where JUL is pointing. That GLO presents the word more slowly the second time around may be a way to make it more accessible to him with the aim perhaps for him to repeat it, but he does not. GLO's production is therefore a type of technique that may allow novices to better apprehend the sounds of their input language to subsequently facilitate their use. The image[6] below (Figure 3) shows JUL pointing at the "mosca" (fly). It also shows GLO just after looking at him as she produced the word "mosca" (fly) a second time and as she looks in the direction of JUL's finger.

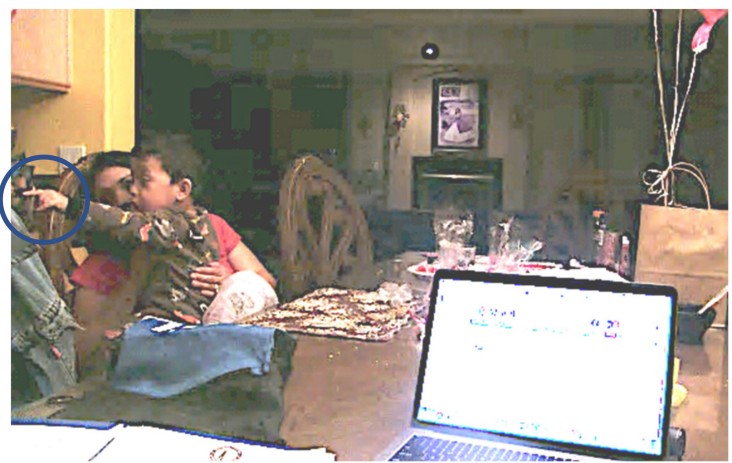

**Figure 3.** JUL's multimodal production to reference a "mosca" (fly).

So far, ERI has not said much. He seems to be unratified but observing the dyadic encounter in front of him. However, following GLO and JUL's brief exchange, ERI zooms and adjusts the camera asking LIN (l.5) from across the dining room table what she is doing. This is perhaps a way to ratify both himself and LIN into GLO and JUL's dyadic interaction. Or maybe to initiate a dyad with LIN on another topic. Nevertheless, when ERI calls her, LIN repeats the OH Spanish word "mosca" (fly), peering up from behind the couch in the living room where she has been quietly playing alone. The following figure (Figure 4) depicts LIN looking up and over the couch towards GLO and JUL's dyadic participation framework. Through LIN's Spanish repetition of OH Spanish, she becomes a ratified member, subsequently reconfiguring the social encounter into a triad.

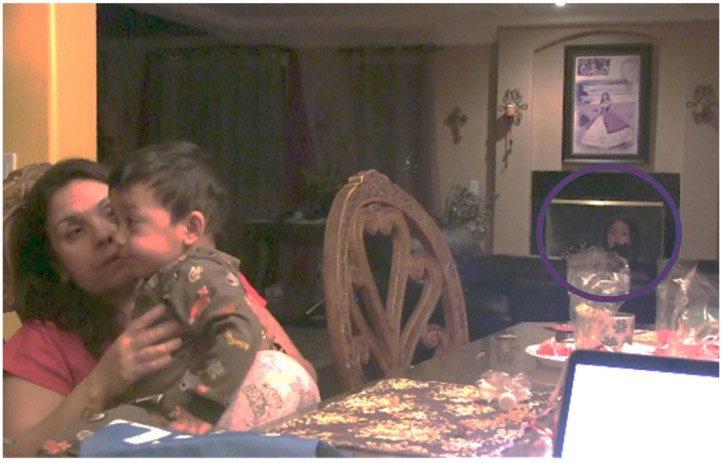

**Figure 4.** LIN captures and repeats the overheard Spanish word "mosca" (fly) as she looks over the couch.

LIN answers ERI (l.6), seemingly more interested in repeating the OH word "mosca" (fly) again, which she does only a couple of seconds later but this time with rising intonation (l.8) as if asking a question. While it is difficult to determine who she addresses, a strong guess is that she interrogates GLO regarding the "mosca" (fly). Indeed, GLO introduced the word into the interaction with JUL. Furthermore, LIN's question overlaps with GLO's statement (l.9) where she tells JUL in Spanish "es una mosca" (it's a fly) and in doing so, she expands her original utterance from one to three Spanish words, thus allowing for richer, though not overwhelming CD Spanish input. Moreover, this may be interpreted as a Repetition Strategy on GLO's behalf, a discourse strategy that does not explicitly require the child to respond, although they may and in the proposed target language. It thus aligns with a monolingual discourse strategy that aims at keeping the conversation in one language. What happens next is extraordinary. Despite GLO's efforts to encourage JUL to repeat the word "mosca" (fly), LIN (l.10), by language mixing, encourages him to say it too. Even if LIN is not an adult, has limited productive ability in heritage Spanish, and is not the central language teacher per se, her utterance may be likened to Lanza's (1997) Language Switch Strategy. That is, even if JUL does not seem to produce the target word, LIN language mixes, which is a bilingual discourse strategy that instead opens a space for the interaction to be carried on in Spanish and/or English. This type of linguistic behavior is typical in bilingual and bicultural households since "The home is a 'safe space' for bilingual families" (Kiaer 2023, p. 9) where children can re-create and explore through their languages. Nevertheless, LIN goes back to playing on her own, and the triadic interaction goes back to a dyad as quickly as it began.

This sequence illustrates how language mixing and dilingual conversations (Saville-Troike 1987; De Houwer 2009) are acceptable and typical forms of interaction among the bilingual, bicultural, and multigenerational family members in this *tercera Hispanidad*. A dilingual interaction refers to the negotiation of meaning without the use of a shared code, i.e., the same language. It shows how LIN takes on the role of scaffolding expert in Spanish to which she has been socialized through CD speech. Her ability to participate so actively in her brother's socialization to Spanish indicates that she has internalized her multilingual input, accepts to fully participate in this type of "languaging" (Linell 2009; Chèvrefils et al. 2023), and feels empowered to become an agent of language socialization herself.

*4.3. Chile*

This last analysis (Box 3) builds on the previous two where LIN attunes to OH Spanish input to practice and to engage in conversation. The participation framework is first dyadic, then triadic, before becoming a multiparty encounter. However, unlike in the previous examples, even if the adults are all bilingual speakers, here they represent three generations. This extract is also special because it is one of the rare occasions where LIN's father MAR participates in the unfolding encounter. Moreover, unlike the first example that is dominated by Spanish discourse, the multiparty interaction here is dominated by language mixing by the adults. In the following analysis, LIN uses the OH Spanish input despite the abundance of language mixing to which she is constantly exposed. LIN overhears ROX say the word "chile" (chili). ROX's comment prompts LIN to repeat the OH word. The interaction takes place in January 2019. Only MAR is sitting at the dinner table at first. ROX is standing near MAR facing him. GLO is standing on the other side of MAR and is partially visible. LIN and JUL are playing in the living room, but they eventually make their way to the dinner table. ERI is sitting in the living room, completely removed from the interaction. The camera is placed on a tripod near a corner of the table. The main participants (MAR, ROX, LIN, and JUL) are visible so their gaze and gestures are analyzed. The discussion is unfocused. GLO is an unratified overhearer and is shuffling through mail. She addresses ROX from time to time. MAR and ROX are the ratified speakers. They are discussing the ongoing teacher strikes and LIN's attendance at school. Finally, LIN and JUL are unratified overhearers until JUL approaches MAR.

**Box 3.** LIN is 4;9 (*Plurilingual Transcript 3; ex. 21: JAN_09_2019_LIN_GLO_ERI_ROX_MAR_chile*).

---

1. *ROX: cause [: because] I don't want her not to have perfect attendance. @MAR
2. *ROX: so +... @MAR
3. *MAR: hey where is Mario and Leida doing it at um Incredibles? @ROX
4. *ROX: at I think it's um +... @MAR
5. *ROX: [- mix] la hijastra de Denis@s. @MAR
   　　　　　　　　　Denis' stepdaughter.
6. *JUL: Daddy.
7. *LIN: can I go Incredible? @MAR
   　　　(LIN joins adult interaction, standing on chair next to MAR, and looking at him from
   very close distance)
8. *JUL: xxx. @MAR
9. *LIN: can I go Incredibles? @MAR
10. *MAR: what's up? @JUL
11. *JUL: that. @MAR
   　　　　　(Gazes and points at drink on table next to MAR's bowl of soup)
12. *LIN: hey Daddy what you say?
13. *MAR: you want some? @JUL
14. *ROX: on your birthday. @LIN
   　　　　　(Gazes at MAR and JUL)
15. *JUL: yeah. @MAR
   　　　　(Gaze goes from drink to bowl of soup)
16. *ROX: [- mix] no it has chile@s. @MAR
   　　　　　　　　No it has chili.
17. *LIN: [- spa] chile! @JUL
   　　　　　　　　Chili! (Brings index finger and thumb briefly to mouth, then leans toward
   MAR pointing at his chest with right arm extended, but looking down at bowl)
18. *ROX: no no.
   　　　(Looking over all of them)
19. *JUL: [- spa] chi(le) chile. @MAR
   　　　　　　　Chili chili.
20. *LIN: (Looks at JUL, stretches right arm again to block spoon with her hand)
21. *ROX: [- spa] chile. @JUL
   　　　　　　　Chili. (Looks at JUL)
22. *MAR: [- spa] poquito. @JUL
   　　　　　　　A little bit. (Maintains his gaze on JUL)
23. *JUL: no. @MAR
   　　　(Removes himself from spoons trajectory by moving to MAR's left)
24. *GLO: [- spa] tiraste +... @ROX
   　　　　　Did you throw +...
25. *LIN: [- spa] chile!
   　　　　　　　Chili! (Looks at ROX)
26. *JUL: LIN help me.
27. *ROX: I didn't throw nothing. @GLO
28. *GLO: [- spa] no encuentro +...
   　　　　　　　I can't find +...
29. *ROX: LIN move from there you have to shower girl.

---

As the school strikes discussion continues, in their dyad ROX (l.1) comments to MAR that she prefers LIN to have perfect attendance. This is very important to them both. Their preoccupation with education is an aspect that they value, discuss often, and try to instill in LIN. Indeed, it has been argued that "Through participation in everyday routines and social interactions as both active participants and observers, children are socialized into culturally specific orientations towards work, education, time, morality, responsibility, individualism, success, well-being, and what it means to be a family" (Paugh 2008, p. 105). In the interactional frame, LIN is an overhearer; however, she may be close enough to gather through her parents discourse an understanding of what is expected of her in terms of education. Next, LIN joins the adult interaction (l.7), forming a multiparty participation framework. She does so multimodally first by asking her father if she can go to a party that

ROX and MAR were discussing and by getting on the chair next to MAR, looking at him from a very close distance until they are close enough to exchange an affectionate kiss. The figure below (Figure 5) seizes the moment when LIN, after asking her father for permission to attend the pizza party, moves in for a kiss.

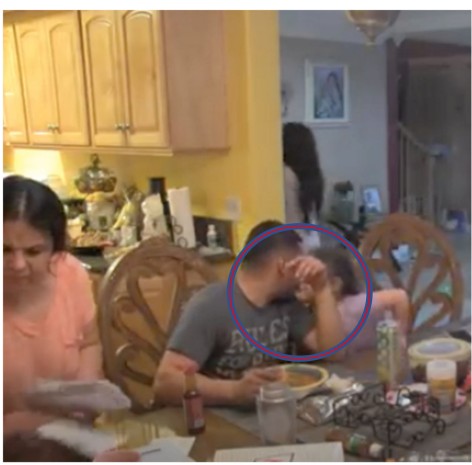

**Figure 5.** LIN enters interaction multimodally with her father MAR through speech and affection.

Two points may be highlighted here. First, LIN's reaction to the OH conversation between her bilingual parents shows that even if she is busy playing with JUL, she is aware of what is said around her. This may be especially true when what is said is in English, her dominant language. Second, at this point, the dyadic interaction between MAR and ROX has expanded to a multiparty participation framework of four. This new multimodal and bilingual frame subsequently engenders gestures and the use of Spanish by LIN and by JUL. JUL (l.11) gazes at and points to a drink on the table next to MAR's bowl of soup and asks for it. After realizing that JUL asks for something, MAR asks JUL if he wants some of his soup (l.13). ROX who is partially visible and with her back turned to the camera returns to the interaction, although she was still overhearing. She stands behind LIN's chair, gazing at MAR and JUL who are both to her right. JUL (l.15) agrees with MAR's proposition, his gaze going from the drink to the bowl of soup. However, ROX (l.16) language mixes to suggest that it is not a good idea because it has "chile" (chili). In this case, the English word resembles the Spanish one but is slightly different, as not only is the phonology[7] of the production Spanish, but so is the term itself. It is not clear whether she addresses MAR so that he does not feed JUL, or if she addresses JUL so that he does not accept it from MAR, or both. A strong guess is that ROX addresses JUL since the adults frequently advise the children when something is too spicy, and it is up to them to decide whether they want to try it. Nevertheless, ROX's language mixing is a source of OH speech for LIN from which she extracts the Spanish word "chile" (chili). LIN, who is still sitting on the table facing MAR, takes her right hand in the form of a fist, except for her index finger and thumb, briefly bringing it to her mouth. Through her representational gesture, she shows that "chile" (chili) is spicy and should be consumed with caution. The following figure (Figure 6) depicts the moment LIN brings her finger to her mouth after overhearing that the soup is spicy.

Furthermore, LIN's gesture precedes her repetition of the OH Spanish word. The gesture (l.17) is followed by her enthusiastic Spanish utterance "chile!" (chili), perhaps a warning addressed to JUL following ROX. As she does so, LIN leans towards MAR pointing at the center of his chest with her right arm extended, although she is looking down at the bowl. The figure below (Figure 7) shows LIN deploying her linguistic resources multimodally. LIN shouts the OH word "chile" (chili) in Spanish, as she energetically points in her father MAR's direction.

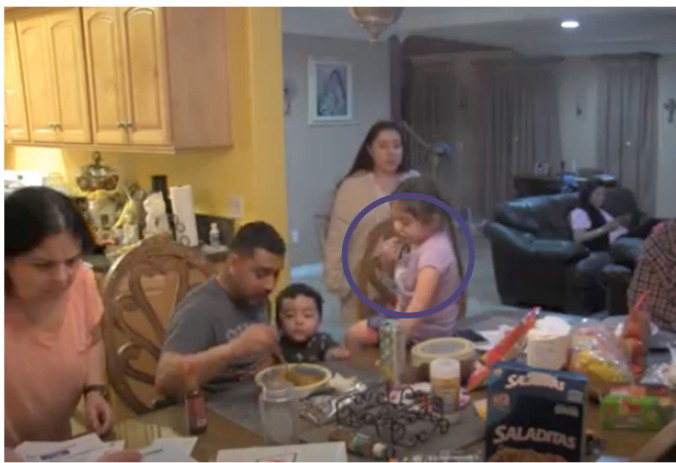

**Figure 6.** LIN takes her fingers to her mouth after overhearing her mother ROX say the word "chile" (chili).

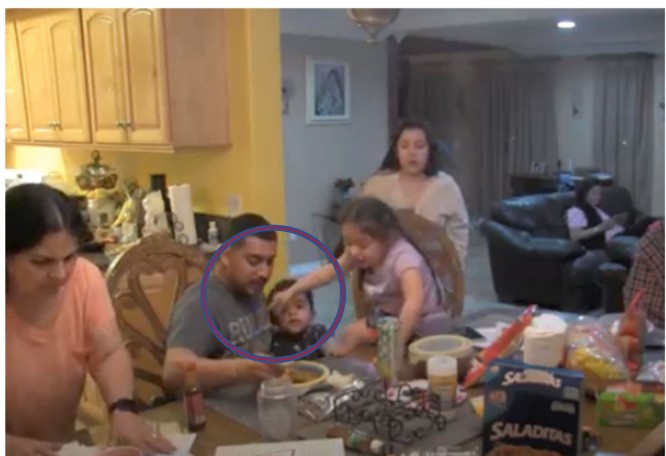

**Figure 7.** Multimodally LIN repeats the OH word "chile" (chili) and points in MAR's direction.

Moreover, there seems to be a sort of discordance between what LIN is pointing at (MAR) and what she is looking at (the soup). It may be interpreted as a way for her to indicate to her little brother JUL that her father is trying to feed him something spicy. Through her bilingual and multimodal interaction, she is referencing both MAR (by pointing at him) and the soup (by saying "chile" (chili) while looking at it). Then, ROX, who is still looking over all of them, says no again twice (l.18) when MAR brings a spoon of soup to JUL's mouth. Furthermore, JUL (l.19) repeats "chile" (chili) in Spanish as he gets ready to take it. At the same time, LIN (l.20) looks at JUL, stretching her right arm again to block the spoon with her hand from getting into JUL's mouth. ROX (l.21) insists again while looking at JUL and repeats "chile" (chili), but MAR responds in Spanish, suggesting that JUL try just a little bit (l.22), maintaining his gaze on JUL. Much like when ROX said "no" further up, here again we do not know whether MAR addresses ROX or JUL in Spanish. It appears, however, as though the language of previous utterance, or Spanish, may have influenced MAR's language choice. Up until now, MAR has only addressed ROX and JUL in English, but this time he addresses one (or both?) of them in Spanish. Nevertheless, before MAR can put the spoon into JUL's mouth, JUL removes himself from its trajectory by moving to the left of MAR (l.23). JUL refuses to eat the soup, so MAR continues eating. LIN (l.25) repeats the OH Spanish word "chile" (chili) again in a playful tone, seemingly looking at ROX, but it is difficult to tell since her back is now partially turned to the camera. The interaction ends as LIN stands on the chair and turns around to see what ROX is doing.

In this sequence, LIN displays her multilingual and multimodal languaging skills, and her behavior also illustrates how much she feels empowered to align with the scaffolding

community of experts in relation to her more novice brother. She has been socialized to her expertise in the characteristics of Mexican food and can produce the multilingual and behavioral cultural practices used with children in her community.

## 5. Discussion

Our discussion creates synergies between the quantitative and the qualitative analyses, framing them around the family's often bilingual, typically multiparty, and multigenerational ecology. In doing so, we answer our primary research question which asks: *how does overheard Spanish input impact a child's word learning in a third-generation heritage Spanish acquisition and socialization context?* Our secondary question asks: how much are Spanish, English, and language mixing spoken in the environment? The discussion underscores the role of child attention, agency, and empowerment in using OH Spanish in particular, but also in language development in general. We also show that using a mixed method is valuable since our detailed analyses captured communicative features that were not included in our quantitative analyses, especially the role of gesture in multilingual, multigenerational family interaction. Indeed, detailed descriptions of the sociolinguistic context of bilingual experiences are rare. For example, Surrain and Luk (2017) found that only 30% of 186 studies that compared bilingual and monolingual labels and descriptions described the bilingual setting. Finally, our discussion addresses how our study filled the three gaps outlined in our introduction, namely, if similarities or differences emerged that have been observed in other studies on OH input and word learning.

There are correspondences between the family's linguistic soundscape, their individual language presentations, LIN's use of heritage Spanish words, according to their semantic domains, and our thick descriptions that underscored the family's dynamic language practices. At least three issues inherently related to OH input and lexical development may be raised. The first issue is the finding that Spanish use significantly dropped longitudinally. The second one concerns the results showing that almost all bilingual speakers used English the most, followed by Spanish, and lastly language mixing. The third issue is in reference to the finding that LIN's most robust semantic domain in our data, in terms of Spanish word use, was related to "Food". The results portray a heritage bilingual ecology where English at the family and the individual level eclipses Spanish. Nevertheless, the minority language, Spanish, is present in this third-generation setting. Spanish is transmitted to LIN perhaps not predominantly through CD speech, but rather through what she gathers through OH speech. As shown in Alvarez (2020, 2023) and presented in LIN's socio-linguistic profile, LIN's main source of input is CD English. However, with relation to OH Spanish, this type of input is omnipresent when it comes to food talk or during dinner time activities more generally. LIN is also more prone to use Spanish when she is an overhearer in a participation framework. This means that as an overhearer, she is responsive to the OH Spanish input in joint attentional frames, allowing her to exercise her agency and use OH Spanish to engage in interaction which also assists her in developing bilingual competence. It seems particularly efficient, for example, when she understands that her little brother is being socialized to specific cultural practices. The under-explored role of OH speech in multiparty, multigenerational settings may therefore be a powerful vector for lexical development in heritage bilingual ecologies, even if OH input alone might not be sufficient for children to become proficient users of the heritage language.

As we have shown in our detailed analyses, in considering the OH Spanish word "tortilla" (flatbread), LIN, while in the presence of multiple bilingual family members, shows that she is attuned to, and through her agency, uses the OH Spanish input to practice using the term. In describing the OH Spanish word "chile" (chili), LIN is highly perceptive to OH Spanish input even when it is presented through language mixing. Finally, our analysis of the OH Spanish word "mosca" (fly) shows that LIN is attentive to the OH Spanish despite her spatial distance from the joint attentional interaction that is unfolding out of her sight at first. A common thread in these three examples is that, in an environment where Spanish use significantly decreases while English use rises longitudinally, LIN

nevertheless uses some heritage Spanish, and with the adequate phonology. Indeed, her Spanish production at the word level is native, or near-native, in line with research suggesting that phonology is the most resilient domain in heritage speakers (Montrul 2016), even though she is English dominant. She especially tends to do so when she is not asked to speak Spanish. That is, when the input is directed at a third party in a joint interactional frame, in this case her little brother JUL, or when the learning situation is not so didactic. Attention to OH Spanish input and child agency therefore plays a critical role in LIN's lexical development or her use of Spanish words, especially concerning food. OH talk related to food, like "tortilla" (flatbread) or "chile" (chili), or other everyday nouns, like "mosca" (fly) and other cultural objects, supports the construction of not only her bilingual and bicultural identity, but also the development in bilingual competence in such a multilingual context (Blackledge and Pavlenko 2001) that harbors rich linguistic diversity (Wigdorowitz et al. 2020, 2022). LIN's Spanish lexical development is also supported by the multigenerational adults' attentiveness to child speech. For example, when ROX overhears LIN repeat the Spanish words directed to JUL, ROX, through her agency, repeats the OH Spanish words to LIN's benefit. This is a situation where LIN speaks "the right language in the right circumstances" (De Houwer 2009, p. 133), an inherent facet of the bilingual language acquisition and socialization process, but also a moment when LIN, through her agency, impacts the course of the interaction.

Our examples showed a range of multigenerational bilingual speakers in interaction who mostly privilege the use of English (Alvarez 2020, 2023) in this L.A.-based family. Yet, this extended-family ecology engenders social encounters where the language of conversation may be entirely in Spanish, especially in the presence of GRC and GRT (both 1G), and English or language mixing in the presence of GLO (1.5G), ROX, and ALE (2G). The second-generation bilingual speakers typically refrain from using English or language mixing in the presence of GRC and GRT, whose English is clearly not proficient. Such a bilingual identity thus requires sensitivity to the linguistic needs of one's interlocutors. Nevertheless, throughout our longitudinal data collection, LIN's use of heritage Spanish rarely extends beyond the one-word stage, as attested in the semantic domain analyses, even when the multigenerational bilingual speakers in the interaction only use Spanish. However, LIN is attuned to, and through her motivation, she is empowered to use OH input to practice Spanish, to engage in multiparty interaction, and to encourage other third-generation novices to speak Spanish. For example, in considering parental discourse strategies (Lanza 1997; De Houwer and Nakamura 2021) intended to create mono- or bilingual language learning environments, we learn that parental discourse strategies are not exclusive to the adults. When GLO deploys a monolingual discourse strategy (Repetition) to encourage JUL to use heritage Spanish, LIN is listening, and through her agency in the participation framework, first as an overhearer and then as a ratified participant, she deploys a bilingual discourse strategy (Language Switch). GLO and LIN's discourse strategies seem to oppose each other on the surface. But as multigenerational speakers, they both implicitly aim at having JUL repeat the Spanish word by rendering the language learning environment monolingual Spanish (GLO) and bilingual (LIN). Language mixing, though it is the least used mode of communication in the family, may thus support heritage Spanish use. It may also allow for socialization into various aspects of the family's Mexican culture. However, it raises the question of socialization into language mixing and whether it truly opens a path towards robust heritage Spanish lexical development, and thus productive heritage bilingual speakers.

As we have shown in our detailed analyses of three sequences, in these multiparty, multigenerational encounters, successful interactions are also supported by using gesture to aid in the bilingual and co-constructed meaning making process. This is a crucial social-interactional aspect that was not captured in our quantitative analyses focused on speech and on comparing the use of each specific language. For example, in the "chile" (chili) extract, LIN uses heritage Spanish in multimodal interaction (pointing, gaze, and blocking). LIN brings her finger to her mouth when she overhears the word "chile" (chili), but also points at the spicy soup to warn her little brother JUL that it is too spicy. In the "mosca"

(fly) example, pointing and gaze are also used to designate the object in the joint attentional frame. Nevertheless, one of the difficulties in these multigenerational, multiparty frameworks is that the speakers' gaze does not always seem to point directly at the person they are addressing. It is therefore likely that some utterances are destined to more than one participant in these types of encounters, indicating that the participation framework is dynamically flexible, inclusive, and allows the child to engage in the conversation at all moments. This also creates a bilingual ecology where enhanced attention skills are important to partake in the flow of the conversation. The frontiers between CD and OH speech are thus extremely permeable, enabling children to join the participation framework at will and thus express their own agency. The category OH speech could thus be conceptualized on a continuum with CD speech, and it is liable to reinterpretation in the family language socialization practices.

Finally, we turn to how our study filled the three gaps outlined in our introduction. The first gap regards the lack of research on heritage bilingual development grounded on the language socialization paradigm (Ochs and Schieffelin 2011). Our framework provides not only quantitative per se, but also qualitative analyses of the dynamic linguistic practices of a heritage Spanish bilingual community in L.A. Joint interaction in less didactic contexts are thought to assist children in their vocabulary development. However, unlike what Bloom (1998) advanced, in the present study, less didactic interactions do not seem to accelerate LIN's heritage Spanish development. At best, these informal settings may help LIN maintain some Spanish, especially since she is sensitive to OH input, and her agency allows her to be selective in co-constructing her bilingual and bicultural knowledge. Nevertheless, in line with Orena et al. (2020), our study anchored on the language socialization paradigm revealed the important shifts in proportions of the English, Spanish, and language mixed input in this bilingual ecology. Furthermore, our detailed analyses showed that attention to OH Spanish input drives the child's agency, and thus engagement in interaction. The issue of the understudied role of OH input was also a gap we sought to address. OH Spanish allows the family's Mexican linguistic and cultural heritage to continue spatially and temporally despite the dominant pressure of English. But a key ingredient is an attentive, motivated child willing to engage in interaction if this is permitted in their culture. There thus exists a strong link between OH input, word learning, attention, and agency in interaction. This finding supports interdisciplinary research (Akhtar et al. 2001; Akhtar 2005; Floor and Akhtar 2006; Gampe et al. 2012). The last gap we aimed to address was related to age. Word-learning and OH input research has largely centered on children around two years old (Foster and Hund 2011). Our contribution examines word learning and OH speech in a child between 3;10–4;9. Heeding Boderé and Jaspaert's (2016) recommendation, we provide a picture of the language acquisition and socialization of an older, more linguistically productive child, leading to richer interactive analyses overall. Indeed, interactional descriptions of OH input with older children are still under-explored in heritage bilingual contexts. Our paper thus presents a much-needed contribution in child bilingual language development research.

Despite the dynamic use of Spanish and language mixing in interaction, the most frequent mode of communication at the family and the individual level is English, as we have shown in our quantitative analyses. However, LIN uses OH Spanish words on her own. LIN is keenly aware of what we have so far categorized as OH speech, as well as to the bilingual adults' rich language practices. LIN may thus be implicitly socialized to answer in English when she is spoken to in Spanish regardless of where her interlocutors are on the bilingual continuum. Only further research in more varied interactional contexts, with other multigenerational speakers, and embedded in multiparty frameworks, may reveal if these types of implicit linguistic practices promote lexical development, and thus acquisition of and socialization into heritage Spanish as a third-generation child in L.A. As Hoyle and Adger (1998) advance, children, as they go about their lives, are immersed in a broad range of social settings which in turn expand their social networks. Children thus are empowered; engendering novel uses within their linguistic repertoire:

As part of this expanding social life, children's peer, sibling, and play interactions constitute a rich site of language socialization. The language varieties and styles that children use in peer and sibling interactions and the particular ways they deploy their linguistic repertoire, strongly impact and may have profound implications for language change, maintenance, and shift (Howard 2008, p. 193).

## 6. Conclusions

Fleeting moments such as those we have analyzed in our examples may not be enough to become proficient speakers of heritage Spanish. However, bilingual discourse strategies allow us to consider the issue of bilingual socialization. It would appear as though a language environment where language mixing is allowed both favors the continued use of the two languages and encourages dilingual conversations that facilitate multigenerational communication (Chung 2010) and fosters bilingualism (Gorter 2013).

Apprehending individual bilingual language experiences remains an intricate enterprise (Luk and Bialystok 2013; Silva-Corvalán and Treffers-Daller 2015; Köpke and Genevska-Hanke 2018). Thanks to the use of both quantitative and qualitative analyses, the present paper has highlighted that what we have coded as the OH input, Spanish in this case, to which children are exposed in their communities of practice, is an important source of linguistic knowledge. In the present case study, we focused on the enrichment of the child's lexicon in the heritage language which is used minimally by the child and in the input explicitly addressed to her. OH input also serves to scaffold young children's bilingual and bicultural construction in interaction as practices are permeable, flexible, inclusive, and the participation frameworks are dynamically constructed by participants' engagement in the conversation. Through the adults' language choices and use that are often implicit and non-goal oriented, children acquire and are socialized into the particular modes of communication, as well as into the distinct language varieties of their entourage, for example, the Los Angeles variety of Spanish (Parodi 2011) in the present case. Certainly, while the quantity of Spanish production is small, LIN is nevertheless actively developing bilingual competence through her attention, agency, and empowerment as evidenced in her initiations of OH Spanish. Of course, LIN is also acquiring and socialized into the use of English, language mixing, and a receptive form of bilingualism in Spanish. These language practices, including gesture, gaze, and facial expressions, facilitate meaning-making and help maintain the use of heritage Spanish in this L.A.-based multigenerational family.

Substantial quantitative and detailed qualitative analyses are needed to demonstrate the importance of dynamic participation frameworks in multiparty interactions and in multilingual contexts with high contextual linguistic diversity. Our case study contributes to the ongoing discussion of what it means to be bilingual in general, providing definitions of heritage bilingualism in particular. In addressing the Special Issue's key question, we add to our limited understanding how linguistic diversity and OH language use in a tension-ridden bilingual contact zone impacts the outcomes of heritage bilingual acquisition and socialization. The results underscore the role of what is categorized as OH Spanish and lexical development in L.A., showing that heritage bilinguals do not need to be proficient in the minority language to be impacted by their presence in society (Wigdorowitz et al. 2020, 2022). Our results may even suggest that heritage bilingualism is not just about abstract grammatical rules, divergent, or incomplete acquisition (Montrul 2016) since significant variations in language use and exposure exist in bilingual communities. Thus, following a social justice perspective (Ortega 2014, 2019), heritage bilinguals in general and heritage Spanish speakers in this study inherit bilingual and bicultural practices constructed by, with, and for their community of practice.

**Author Contributions:** Conceptualization, E.A. and A.M.; methodology, E.A.; software, E.A.; validation, A.M.; formal analysis, E.A. and A.M.; investigation, E.A.; resources, E.A.; data curation, E.A.; writing—original draft preparation, E.A.; writing—review and editing, E.A. and A.M.; visualization, E.A. and A.M.; supervision, A.M.; project administration, E.A. and A.M.; funding acquisition, E.A. and A.M.; All authors have read and agreed to the published version of the manuscript.

**Funding:** The data collection was funded both by Université Sorbonne Nouvelle through a three-year doctoral contract, and by Labex Empirical Foundations of Linguistics (EFL) through a mobility grant.

**Institutional Review Board Statement:** This study was conducted according to the guidance of the Declaration of Helsinki, and approved by the Ethics Committee of the doctoral school in which this study was conducted at Université Sorbonne Nouvelle.

**Informed Consent Statement:** Informed Consent was obtained from all participants in this study.

**Data Availability Statement:** The data supporting the conclusions in this study are available on request from the corresponding author (accurately indicate status).

**Conflicts of Interest:** The authors declare no conflicts of interest.

## Notes

[1] This study is derived from the results of the Ph.D. dissertation titled "Third-generation heritage Spanish acquisition and socialization in Los Angeles, California. *A cognitive-functional and socio-interactional mixed methods case study of Spanish-English bilingualism".* It was defended at Université Sorbonne Nouvelle in January 2023.

[2] These figures are based on data that was collected in February 2016, analyzed, and presented within the framework of my "M2 Recherche" dissertation. However, it is important to highlight that the research methodology, namely the transcription was carried out exclusively on Excel as opposed to CLAN as is presently the case. Therefore, while it is evident that clear-cut comparisons are not entirely possible for methodology related reasons, the figures presented may help give a general overview into LIN's dynamic bilingual practices when she was 1;10.

[3] We excluded two coding categories related to the metadata. The Line Number and the Media File do not have a baring on the results of the present case study.

[4] This may be likened to the French who often accompany their meals with sliced pieces of baguette on the table.

[5] "Tortilla" (flatbread) may be considered a language-neutral word since it works in both languages (De Houwer 2009). However, Spanish plosives /p,t,k/ are unaspirated and the family is sensitive to this and other phonological aspects as seen in Alvarez (2023). In the example above, the word initial /t/ in "tortilla" (flatbread) is unaspirated. It is thus coded as Spanish since the English version of the term would be aspirated. Along with aspirated plosives, other distinctive features of Spanish words marked with English phonology include difficulty producing the Spanish trill, producing the alveolar lateral approximant in English etc.

[6] The following two images have been formatted as best as possible to better see JUL, GLO, and LIN in interaction.

[7] Like the "tortilla" (flatbread) example, "chile" (chili) could be considered language-neutral. However, unlike "tortilla" (flatbread) graphically, chili is spelled differently. Furthermore, in phonological terms the alveolar lateral approximant /l/ is not produced the same in Spanish and English. Indeed, producing the Spanish /l/ requires a flat bodied tongue, whereas producing the English /l/ requires a raised tongue (https://uw.pressbooks.pub/jorgestextbook/chapter/3-8-the-letter-l/ accessed on 6 March 2024). In the example above, LIN produces the word "chile" (chile) with Spanish phonology, marked by her realization of the /l/. Another indicator that the word is produced in Spanish is that LIN produces the vowel-final sound as /e/. Had the word been produced in English the vowel-final sound would be produced as /i/.

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
