# Peer review of "Third-Generation Heritage Spanish Acquisition and Socialization: Word Learning and Overheard Input in an L.A.-Based Mexican Family"

_languages, doi:10.3390/languages9030108_

Round 1
Reviewer 1 Report
Comments and Suggestions for Authors
Author Response
Dear Reviewer,
Thank you kindly for your feedback. Attached you will find my responses.

Reviewer 2 Report
Comments and Suggestions for Authors
The paper analyzes the role of overheard (OH) input in the socialization and acquisition of heritage Spanish using a longitudinal case study of a third-generation L.A.-based Mexican family. From a usage-based perspective, the paper is guided by a clear research question: how OH input in linguistic interaction and socialization settings motivates a third-generation child to use words in her heritage Spanish. More specifically, it shows the importance of both the dynamics of attention that emerge between active and passive participants in linguistic interactions and the agency of the child.
The added value of the contribution lies at least in two aspects: on the one hand, the role of OH input in the socialization of heritage speakers has been insufficiently studied; on the other hand, the field of study of heritage languages can benefit from the approach of interaction ethnography.
Methodologically, the study is well-structured and contains empirical material adequate for the objectives, both in terms of the observations and the spatial-temporal dimension: the home, more specifically a kitchen meeting environment at three moments during the period of one year, crucial for the linguistic development of the heritage speaker studied (ages between 3;10 to 4;10). In addition, the authors provide an approach that combines quantitative and qualitative methods, something that can benefit the study of heritage languages, especially considering the complexity of the phenomenon. The quantitative description of the soundscape of the family, the linguistic profiles of the members and the main semantic domains of heritage language use serve both as a justification for the selection of the interactions to be analyzed and as an analytical tool for the ethnographic description.
From the point of view of the special issue "Language Use, Processing and Acquisition in Multilingual Contexts", the proposed study conforms to several aspects proposed by the editors:
- it presents the multiparty interactional context in the linguistic socialization of a heritage speaker in a multilingual English-Spanish environment with speakers of different generations with different profiles.
- the paper's focus on OH input makes a useful contribution to the study of linguistic exposure in multilingual contexts by describing the sociolinguistic environment of bilingual experience in socialization setting, especially from the perspective of the multimodality of interactional contexts.
However, I identify several important aspects that should be addressed.
In relation to the contextualization of Spanish-English bilingualism in the United States, discussed by the author in the introduction, it would be better to define the tercera Hispanidad based on key demolinguistic and educational data to understand the conditions for the intergenerational transmission of Spanish.
Some structural, methodological or conceptual weaknesses should be solved:
- Page 2 introduces 3 gaps in the field of study in which the paper is framed. However, the first one does not describe any gap but simply presents the linguistic socialization paradigm as the basis of the analysis.
- In the section on materials and methods, the titles and frames of the subsections do not completely correspond to what is presented in them. Subsection “2.1. Data” describes the participants (the family members) and the material (the recordings); subsection “2.2. Methodology” presents the data gathering but does not describe the method. In addition, there is a lack of discussion of three aspects that define the methodology used in this work: the longitudinal nature of the data, the mixed methods approach, and the variables or factors subject to observation as well as the assumptions on which the analysis is built. Finally, I understand that the coding question is an important aspect of the previous work on which this research is based; however, I believe that in this paper it is not relevant enough to give it so much weight. I would recommend moving the table of categories to an appendix.
- The methodological part should describe, at least minimally, some of the tools and methods used in the analysis, such as soundscape, language presentations and semantic domains.
- Another important methodological aspect: one of the central points of the paper is that the analysis of its three examples shows how joint attentional frames are constructed. Therefore, I believe that the methodological part should address that aspect more thoroughly.
- The concept of bilingual continuum is used in several places in the paper. In the introduction it is presented as referring to the profiles of bilinguals "one may become more or less mono- or bilingual throughout ontogeny" (p. 2). However, in the results it is used to refer to the linguistic interactions of the multilingual family. I believe that the paper would benefit from a broader discussion of the concept, also in relation to the operationalization of the concept: although it speaks of continuum, the corpus of interactions is coded in discrete categories "Spanish", "English" and "mixed". This methodological decision should be discussed in the methodological part.
The paper discusses several aspects that are highly relevant to the study of multilingual contexts of linguistic socialization of heritage speakers in relation to the child's participation in linguistic interaction: attentiveness and agency. The former is a topic that is well covered in the theoretical part where the dynamics of child-directed speech and OH input are presented. However, the paper should give more space to discuss the child's agency, as it is a key concept in the analysis and results. This could be done in the theoretical part and in the discussion. It would be important to include references to current work on child agency in linguistic socialization, such as those found at King, K., & Lanza, E. (2017). Ideology, agency, and imagination in multilingual families. Special issue. International Journal of Bilingualism; or Smith‐Christmas, C. (2022). Using a ‘Family Language Policy’lens to explore the dynamic and relational nature of child agency. Children & Society, 36(3), 354-368.
One last important aspect is that the text should further develop the idea proposed at the end of the paper (lines 761-768). Since this is an important statement that refers to the understanding of heritage speakers and their linguistic development, it should be better connected to the theme of the paper.
Finally, some formal aspects should be addressed:
- In Figure 2 the circles need to be adjusted to the diagram.
- On page 22, in the last paragraph of the discussion section it is not clear if all of it is a quote from Howard 2008. It should be clarified.
- In several places, there is a problem with punctuation at the end of direct quotes, e.g. line 152.
Author Response

(The authors gave the same response as above.)

Reviewer 3 Report
Comments and Suggestions for Authors
This research scrutinizes the dynamics of interactions within a multigenerational bilingual family based in Los Angeles, California. The objective is to examine the role of overheard speech in heritage Spanish in language acquisition by a third-generation child aged 3;10 to 4;9. A specific emphasis is given to the child’s use of Spanish lexical items, mainly to ratify herself into the conversation.
This contribution constitutes a case study derived from video recordings conducted at three distinct temporal intervals spanning 12 months, amounting to a cumulative duration of 24 hours. The authors employ a comprehensive approach, encompassing both quantitative and qualitative analyses. The aim is to elucidate the linguistic landscape within which the child is immersed, intentionally diverting focus from child-directed speech. Instead, the investigation centers on overheard speech and the intricate construction of meaning during dynamic familial interactions.
In my perspective, the study's emphasis on capturing subtle details of communication in multilingual environments aligns seamlessly with the objectives of the Special Issue, as it directs focus towards the intricacies of language use within a multilingual setting. Given these considerations, I support its inclusion in the Special Issue currently in preparation.
I have identified certain points that the author(s) may wish to consider for revision, with the aim of enhancing specific sections of the paper.
ll.16-17: The paper’s stated contribution is somewhat broad and lacks precision. I suggest refining this. It seems that your article aims to capture rather nuanced aspects of communication within multilingual environments. I recommend emphasizing this aspect in your discussion, as it represents the distinctive contribution and added value of your paper.
ll. 69-76: the first of the three listed gaps is not clear for the reader. You might want to rephrase it.
ll.174-177: The observation regarding the presence of significant individual differences is of high relevance and should be incorporated into the literature review at an earlier stage.
l. 183: For enhanced clarity, it is desirable to explicitly highlight in both the abstract and the initial section of the introduction that this study constitutes a case study.
l.263: the section on Coding could benefit from the definition of Language Mixing adopted to code utterances.
Table 1: You mention 14 coding categories, but I can only see 12 in Table 1. Also, I suggest shifting the first column to the left; this could improve readability with respect to the three types of data (Metalinguistic, linguistic, social).
Table 2: Whether the counted utterances exclusively encompass overheard speech or extend to child-directed speech remains ambiguous for the reader. If this only counts overheard speech, the progressive shift to a predominant use of English is not so directly justified by the adjustment to LIN’s dominant language.
The assertion of LIN's dominance in English, introduced for the first time in line 292, prompts questions on the methodology employed for this determination. Given the potential significance of overheard speech in a bilingual child's linguistic development, inclusion of additional contextual details concerning LIN's language exposure and development, such as whether she is a sequential or simultaneous bilingual, the age of entry into kindergarten, and exclusive exposure to English in school, could offer a more comprehensive understanding. The author may consider incorporating a brief linguistic profile of the target child to augment the overall clarity and depth of the study.
I also wonder whether your analysis accounts for additional factors contributing to the family's increased used of English throughout the specified timeframe. It would be prudent to explore variables such as a potential decline in participation from the first generation (who may maintain a stable preference for Spanish), or a more comprehensive adjustment of the entire family unit to the presence of the third generation, extending beyond LIN alone. These factors could potentially influence language dynamics within the family and merit consideration in the interpretation of language shifts over time.
Table 2 exhibits temporal intervals in the columns and languages along the rows, while Table 3 adopts an inverse arrangement with languages in the columns and time frames in the rows. Enhancing the homogeneity of data presentation across both tables could facilitate a more cohesive understanding for the reader. Standardizing the format by maintaining the same orientation for both tables would contribute to a more seamless comprehension of the presented data.
To highlight the stability or variability in language use among primary caregivers over time, one effective approach could be to modify the listing order within the tables. For instance, organizing the columns or rows to first display caregivers with consistent language patterns and subsequently those exhibiting changes can visually emphasize this distinction. In lines 313-322, a more explicit commentary on these observed changes, supported by data from the tables, could be incorporated.
3.3. Semantic domains: The observation that ca. one fourth of the Spanish words used by LIN pertain to food could be directly attributed to the contextual set of the recordings, predominantly conducted during mealtimes (line 233).
The focus on mealtimes during data collection raises concerns regarding the broader meaningfulness of the results. The study might benefit from a more diversified data collection approach, including different contexts beyond mealtimes. This would contribute to a more comprehensive understanding of LIN's Spanish word use across different situations, reducing the risk of results being overly influenced by specific contexts.
Section 4
The detailed and vivid accounts contribute to a richer understanding of the contexts under study. The combination of informativeness and readability enhances the overall quality of the descriptions and conveys the intricacies of the observed phenomena.
Certainly, considering instances where the child ratify herself in English within Spanish conversations could provide valuable insights into her reactions to overheard speech and language preferences. Evaluating these instances would offer a more comprehensive understanding of the bilingual child's dynamics, shedding light on her active participation and choice of language within the context of overheard Spanish interactions.
I also have concerns about the similarity in topic of two of the thick descriptions, particularly because of the prevalence of food-related words borrowed from Spanish to English. A broader range of contexts would not only provide a more comprehensive view of the bilingual child's language usage but also minimize the potential impact of specific themes, such as Spanish food-related terms, on the overall findings.
Ll. 637-638: you write “Spanish is transmitted to LIN perhaps not predominantly through CD speech,…”. Do you have any data on this and how did you come to this conclusion? If data on Direct-Speech are available (maybe through other publications) these could briefly be presented in LIN’s linguistic profiles I mentioned above.
I would also advise authors to explicitly address in the discussion how their study has filled the three gaps outlined in the introduction. Specifically, it would be beneficial to examine whether any notable differences or continuities with studies on overheard speech in younger children have been observed. This clarification would enhance the coherence of the paper and elucidate the unique contributions that the current study brings to the existing body of literature.
Minor points:
l. 43: a full stop is missing after A
l. 65: the full stop after “development” is misleading as it suggests the sentence has come to an end. It makes the sentence reading and parsing difficult.
l. 66: “Alvarez, 2023” is missing in the reference list.
l. 106: should this sub-section be numbered (1.1)?
l. 152: same as for line 65.
Figure 2: I’m sure you already noticed that something went wrong with the figure layout.
Plurilingual Transcript 3: please check spacing between 17 and 18 (right arm extended)
ll. 729-733: Because of the colon above, this seems to be a quote, however it lacks quotation marks and indentation.
Author Response

(The authors gave the same response as above.)

Round 2
Reviewer 1 Report
Comments and Suggestions for Authors
The quality has been improved by removing several sentence fragments. I have not made extensive checks of stylistics on this second revision. The author(s) could use read-aloud functions upon receiving the proofs of this article.
Author Response
Hello, please see the attachment for our point-by-point response to the reviewer's comments.
Best,

Reviewer 2 Report
Comments and Suggestions for Authors
Thank you for the changes and responses. I think the manuscript has improved substantially.
Author Response
Thank you for your thoughtful feedback.